# OpenReview forum: "Aegis: Automated Error Generation and Attribution for Multi-Agent Systems"
_ICLR.cc/2026/Conference — ICLR 2026 Poster_

### Official Review · Reviewer_qv7V · 2025-10-29

**Soundness:** 3
**Presentation:** 4
**Contribution:** 3
**Rating:** 8
**Confidence:** 3

**Summary:**

This work features two major contributions for failure attribution of multi-agent system. The first is a large-scale dataset of annotated failure trajectories (failed agents and error modes), obtained by manipulating successful trajectories. The second is unique training methods (reinforcement learning and contrastive learning) which are made possible thanks to the characteristics of the constructed dataset. Experiments on the test set of the proposed dataset and the OOD Who&When dataset demonstrate the effectiveness of the training data and methods, with SFT being the most effective on average. There are many extra analyses and observations presented, which are of potential value for future works.

**Strengths:**

1. Manipulating successful trajectories is a reasonable and scalable way to create data for failure attribution. This is evidenced by that the resulting dataset has 9000+ samples, which is of significantly larger size than e.g. Who&When dataset.
2. The proposed method of data generation is also controlled enough that it can provide fine-grained information for training methods beyond SFT.
3. Having results on OOD/unseen dataset (Who&When) makes the evaluation more comprehensive and convincing.

**Weaknesses:**

I don't see major weaknesses in the submission.

**Questions:**

1. When validating generated trajectories (i.e., seeing whether the intervention really induces a failure), is there a concrete like rule-based function for that, or llm-as-a-judge is employed? If it's the latter case, how accurate the ground-truth labels are (how much noise is there)?
2. Why GRPO consistently performs worse than SFT (e.g., Qwen3-8B-Thinking + GRPO is worse than Qwen3-8B-Non-Thinking + SFT)? Any theory or thoughts?

---

> ### Author Response · Authors · 2025-11-20
> **Response to Reviewer qv7V**
>
> Thank you for the positive and encouraging review. We appreciate your recognition of the scalability of our data-generation approach, the fine-grained control it provides, and the strength of our results on both in-domain and OOD settings. We are grateful that you found no major weaknesses. We address your questions in detail below.
>
> > Q1: Task validation
>
>   We thank the reviewer for this question about our validation pipeline. Actually, we have already discussed this aspect in **Appendix B.1**. We would like to clarify that the majority of our evaluation methodology employs **rule-based** strategies, with the exception of the GAIA task.  For GAIA evaluation, we use an LLM judge (GPT-4o-mini) to assess semantic correctness.  Given the relatively **short and clear nature of GAIA answers**, the LLM judge can easily determine whether the model's response is incorrect or simply differs in surface-level formatting, such as case, punctuation, or symbols.  This approach is more **robust and accurate compared to exact match evaluation**. Furthermore, recent works, such as **Tongyi DeepResearch** [1], also use LLM-as-a-judge for evaluating GAIA tasks, further supporting the reliability of this method.
>
>   [1] Team, Tongyi DeepResearch, et al. "Tongyi DeepResearch Technical Report." arXiv preprint arXiv:2510.24701 (2025).
>
> > Q2: Performance of SFT and GRPO
>
>   The SFT models consistently achieve a higher average score primarily due to their strong performance on our in-domain test set, Aegis-Bench.  As we analyze in Section 7 and Figure 3c, SFT is highly effective at learning the specific patterns and formats of our data, which can lead to some overfitting and pulls up its overall average on Aegis-Bench and Who&When.  GRPO, in contrast, is not trained to perfectly replicate target strings but to optimize a hierarchical reward function. This encourages the model to learn the principles of good attribution.  The difference is evident in the OOD results: GRPO-trained models achieve better generalization performance on the OOD Who&When benchmark (e.g., 53.12 Agent $\mu$F1 for Qwen3-8B-Thinking + GRPO vs. 45.48 for Qwen3-8B-Non-Thinking + SFT).

---

### Official Review · Reviewer_HRkG · 2025-10-31

**Soundness:** 2
**Presentation:** 3
**Contribution:** 2
**Rating:** 6
**Confidence:** 5

**Summary:**

This paper introduce Aegis, a frame to automatically generate erroneous trajectory for multi-agent system. It's used to construct a large dataset of trajectory with error annotation and modes. The author use the generated dataset to fine-tune LLM to do error attribution with three different learning paradigms. Experiment results show the finetuned model improves on the task of error attribution of multi-agent systems.

**Strengths:**

1. The data generation pipeline seems effective and the constructed dataset could be useful for the community
2. The results show that the constructed data can significantly improve open-source models capability of error attribution
3. The proposed DCL is interesting and looks promising even though it's still lag behind LLMs

**Weaknesses:**

1. The design of the error mode taxonomy is critical, but the authors didn't discuss and reveal how it is done in the main body of the paper
2. The data generation pipeline will result in real positive trajectories and synthetic negative trajectories, this may not align with real world cases where errors are real and diverse

**Questions:**

In addition to the two strategies, using an underperforming LLM for certain steps may be a good way to generate realistic errors but it's not guaranteed to produce error

Also, how does the model performance on aegis-bench correlated with who&when? It would be better to do a correlation analysis to see if aegis-bench is positively correlated with who&when

For DCL, it may be better to use better text encoders, eg, qwen3-embedding for better performance; also it would be good to list the number of parameter of DCL to show it's much smaller compared to LLM

---

> ### Author Response · Authors · 2025-11-20
> **Response to Reviewer HRkG (Part 1/2)**
>
> Thank you for the thoughtful review. We appreciate your recognition of **the usefulness of our data-generation pipeline**, **the improvements enabled by our constructed dataset**, and **the promise of our DCL approach**. We address your concerns in detail below.
>
> > W1: Design of the error mode taxonomy
>
>  We thank the reviewer for this question. We must clarify that our work does not design a new error taxonomy. Instead, we use **the existing, empirically-grounded MAST taxonomy (Cemri et al., 2025)**. As we cite in Section 4.3, the MAST taxonomy was developed through a rigorous Grounded Theory analysis of over 150 real, naturally-occurring MAS failure traces. We adopt this specific taxonomy precisely because it is grounded in real-world data. This ensures the errors injected by Aegis are not arbitrary but are programmatic reproductions of known, realistic failure modes. We believe this is the core reason our model generalizes so effectively to "in the wild" benchmarks like Who&When.
>
>  We appreciate the reviewer pointing this out. As this has been a point of confusion, it is clear our emphasis in the main body was insufficient. We have more explicitly highlighted this choice in the revised paper **(Section 4.3)** to ensure this real-world grounding is clear from the start.
>
>  Cemri, Mert, et al. "Why do multi-agent llm systems fail?." arXiv preprint arXiv:2503.13657 (2025).
>
> > W2: Alignment of Aegis and real-world data
>
> We thank the reviewer for raising this point. We have provided a comprehensive and detailed answer to this in our Global Response, as it is a shared concern among the reviewers. We respectfully refer you to our **Global Response** for the detailed **four-part justification** and the new experimental evidence supporting these claims.
>
> > Q1: Using an underperforming LLM to generate errors
>
>   We thank the reviewer for this insightful suggestion.  We agree that using an underperforming LLM is a valid and interesting approach to generating errors. However, this method presents two major challenges that highlight the unique advantages of our Aegis framework.  First, as the reviewer correctly notes, this approach is not guaranteed to produce a failure.  Second, and more critically, even when it does produce a failure, the resulting trajectory would be unlabeled.  Human annotators would need to perform a costly post-hoc analysis to determine what the root-cause error mode is, which is the exact manual and unscalable bottleneck our paper seeks to eliminate.
>
> > Q2: Correlation analysis of model performance on Aegis-bench with Who&When
>
> We thank the reviewer for this excellent suggestion. We conduct a full correlation analysis across all non-random models to compare performance on Aegis-Bench and Who&When. Here is the full breakdown of Pearson correlation coefficients:
>
> | Metric Granularity | Metric Type | Pearson's r | p-value  | Interpretation      |
> |--|--|---|--|---|
> | Error Type  | Macro-F1    | **0.744**   | < 0.001  | **Strong Positive**  |
> | Agent | Micro-F1    | **0.714**   | < 0.001  | **Strong Positive**  |
> | Error Type | Micro-F1    | **0.606**   | 0.002    | **Strong Positive**  |
> | Pair | Micro-F1    | 0.291       | 0.168    | Weak (Not Sig.)      |
> | Agent | Macro-F1    | 0.166       | 0.439    | Weak (Not Sig.)      |
> | Pair | Macro-F1    | -0.106      | 0.623    | No Correlation       |
>
> The results show a strong, statistically significant positive correlation in Agent Micro-F1 (r=0.714, p<0.001) and Error Macro-F1 (r=0.744, p<0.001). This confirms that the fundamental skills of identifying which agent failed (on a high-frequency, instance-weighted basis) and what type of error occurred are consistent and transferable between the benchmarks.
>
> We also note the weaker correlation for Agent Macro-F1 (r=0.166, p=0.439), which is also expected. This metric is sensitive to rare, 'long-tail' agents, and the specific long-tail agent roles naturally differ between Aegis-Bench (built from our 6 MAS frameworks) and Who&When (some from different sources). The weak pair-level correlation is a function of the task's extreme difficulty. As shown in Table 1 of our paper, Pair-level F1 scores (both Micro and Macro) are substantially lower than Agent or Error scores for all models. These low absolute scores indicate a weak signal and low variance across models, making a stable correlation difficult to establish.
>
> We have added the above analysis and discussion to **Appendix D.2**.

---

> ### Author Response · Authors · 2025-11-20
> **Response to Reviewer HRkG (Part 2/2)**
>
> > Q3: Text encoders and the number of parameter of DCL
>
> Our DCL model is intentionally lightweight and analysis-oriented. We adopt `all-MiniLM-L6-V2` as the shared text encoder (**~23M params**), trained end-to-end with the projection heads and prototypes. Our goal is to probe the effectiveness of the contrastive method itself, rather than have its contribution obscured by the capacity of a massive backbone. The suggested `qwen3-embedding` series models (0.6B-8B) were released in June 2025, which was **after the time when our research began**. Furthermore, swapping in a much larger encoder would have significantly increased the computational cost. We agree that listing the parameter counts is crucial for context. The DCL model is indeed orders of magnitude smaller than the 7B/14B LLMs, as shown in the table we added to the appendix:
>
> | Model                                                                 | #Params                |
> |-----------------------------------------------------------------------|-------------------------|
> | DCL (all-MiniLM-L6-V2 encoder, $\approx$23M; + projection/head/prototypes) | $\sim$23–35M            |
> | Aegis (Qwen2.5-7B)   | 7B    |
> | Aegis (Qwen2.5-14B)         | 14B      |
> | Proprietary LLMs (e.g., GPT-4o-mini, Gemini-2.5) | N/A (proprietary) |

---

### Official Review · Reviewer_nsVH · 2025-10-31

**Soundness:** 3
**Presentation:** 3
**Contribution:** 3
**Rating:** 6
**Confidence:** 5

**Summary:**

This paper presents AEGIS, an automated framework for generating large-scale error attribution data in multi-agent systems built from LLMs. By injecting controlled, context-aware faults into correct trajectories, AEGIS creates over 9,500 annotated failure cases. These are used to train models to improve attribution performance across both synthetic and real-world datasets.

**Strengths:**

1. The work presents a scalable and principled pipeline for generating faulty MAS trajectories with programmatic ground-truth labels. This is a significant contribution, as data scarcity has long been a limiting factor for MAS error analysis.
2. AEGIS supports supervised, RL-based, and contrastive learning methods. The design shows thoughtful alignment between the structure of synthetic data and training objectives, which is rarely done this thoroughly in MAS.
3. The authors benchmark a wide array of models across multiple axes and test both in-domain and OOD generalization. Fine-tuned models on AEGIS surpass many proprietary LLMs in accuracy, showing the power of programmatically generated data for enabling high-fidelity diagnostics. This makes the empirical evaluation strong and convincing.

**Weaknesses:**

1. While the paper proposes a novel Disentangled Contrastive Learning approach, the details of how training pairs are generated remain underspecified. It is unclear how a single correct trajectory is transformed into contrastive pairs and how these are selected. A walkthrough or algorithmic example showing how one trajectory is transformed into training instances for DCL would greatly enhance clarity and reproducibility.
2. AEGIS generates incorrect trajectories by injecting errors into correct ones, but many real-world MAS failures, such as those in Who&When, originate in naturally erroneous decision paths. This creates a potential misalignment between how the data is constructed and how errors manifest in the wild. While the authors demonstrate strong generalization, the conceptual gap between error injection (correct to incorrect) and naturally flawed trajectories (originally incorrect) is not addressed. This weakens the connection to real-world deployment contexts and could affect performance on systems where errors are emergent rather than injected.
3. While the results are strong, the paper would benefit from qualitative examples of common failure modes that the best models still struggle with.

**Questions:**

1. The paper evaluates on the Who&When benchmark, where trajectories can exceed 50 steps. However the model has a maximum input length of 8192 tokens during training. The paper does not clarify how such lengthy interactions are handled. Are steps truncated, summarized, or skipped?
2. Are the injected errors randomly selected, or is there a balancing mechanism to ensure coverage across error modes? Could the model become biased toward more frequently injected errors?

---

> ### Author Response · Authors · 2025-11-20
> **Response to Reviewer nsVH**
>
> Thank you for the thoughtful review. We appreciate your recognition of our **scalable data-generation pipeline, alignment of data and training objectives, and the empirical results**. We address your concerns in detail below.
> > W1: Details about DCL.
>
> We thank the reviewer for this question and for the opportunity to clarify our DCL approach. The reviewer is correct that the successful trajectory is the key. Our DCL process treats each failed trajectory as a **bag** of individual turns. First, a lightweight MIL attention mechanism focuses on the most salient (e.g., Top-3) **evidence turns** in this bag, filtering out noise.
>
> The contrastive pair generation, which is the reviewer's question, happens next. For each of these evidence turns (from the failed trajectory), we define its **positive anchors**: (1) its corresponding "clean" turn from the **original successful trajectory**, and (2) the correct "who" (agent) and "why" (error mode) prototypes from two learnable prototype banks. This anchoring is followed by the contrastive step, where our core Supervised Contrastive Loss ($L_{con}$) "pulls" the evidence turn's representation towards these positive anchors and "pushes" it away from all negative anchors (i.e., all other prototypes and hard-negative samples). This process explicitly shapes the embedding space, forcing the model to learn semantically meaningful representations that distinguish different failure types from their "clean" counterparts.
>
> Finally, a lightweight head combines the "who" and "why" representations to **form a final pair prediction**. This output is constrained by a Hierarchical Consistency Loss ($L_{hier}$) to ensure logical self-consistency (i.e., a pair's probability cannot exceed that of its individual parts), while a standard Classification Loss ($L_{cls}$) ensures overall bag-level accuracy. We have added **Algorithm 1 of DCL and more details to Appendix C.3**.
>
> > W2: Alignment of Aegis and real-world data
>
>   We thank the reviewer for raising this point. We have provided a comprehensive and detailed answer to this in our Global Response, as it is a shared concern among the reviewers. We respectfully refer you to our **Global Response** for the detailed **four-part justification** and the new experimental evidence supporting these claims.
>
> > W3: Qualitative examples
>
>   We appreciate your constructive suggestion. We agree that qualitative analysis is crucial, which is why we included one such case study in our **original submission in Appendix A.1 (Figure 5)**. In that example from the Who&When benchmark, the true root cause was the `Tickets_Pricing_Expert` agent providing incorrect data.
>
>   To further highlight the open challenges that all models still struggle with, we have added a new case study in **Appendix A.2 (Figure 6)**, showing a more subtle failure caused by an incorrect numerical assumption (1 mile = 20 blocks) in the `Verification_Expert`. All models including SOTA baselines and our own Aegis model fail to identify this true root cause, instead blaming downstream symptoms.
>
>   We will include these detailed qualitative analyses in the final camera-ready version, and we plan to expand this line of analysis in future work.
>
> > Q1: Input length during training and evaluation
>
> We thank the reviewer for pointing out this important experimental detail.  The 8192 token limit is used during training, as it is sufficient to cover the vast majority of our generated Aegis trajectories. The reviewer is correct that trajectories in Who&When can be much longer.  To ensure a fair and robust evaluation, we increase the maximum context length to **32,768** tokens during all evaluation runs in our implementation.  This 32k limit is sufficient to cover **over 95%** of the trajectories in the Who&When benchmark without truncation, ensuring the models are assessed on the full interaction history.  This is also well within the operational context window of the models we evaluate, such as the Qwen2.5 series (128k context).
>
> We apologize for omitting this clarification and have added this detail to Appendix D.1.
>
> > Q2: Injected errors selection
>
> We thank the reviewer for this question regarding the data distribution. The error modes for injection are **selected randomly** from the 14 fine-grained failure modes during the generation process.
>
> This is a valid concern, as a highly skewed distribution could bias the model. We provide a full breakdown of the dataset's composition in **Table 2 (Page 18)**. As the table shows, the 14 injected error modes are relatively uniformly distributed across the 24,843 total error instances in the final 9,533 trajectories. For example, the most frequent mode (EM-1.4) has 2,310 instances, while the least frequent (EM-2.5) has 1,513. This relatively balanced distribution ensures that the model receives sufficient training signal across all failure types and mitigates the risk of it becoming heavily biased toward only one or two common error modes.

---

### Official Review · Reviewer_Dj9H · 2025-11-02

**Soundness:** 3
**Presentation:** 2
**Contribution:** 2
**Rating:** 4
**Confidence:** 3

**Summary:**

The paper aims to address the bottleneck of data scarcity in training error-attribution systems for LLM-based multi-agent systems' execution. The authors introduce Aegis, a framework to programatically generate large-scale dataset of annotated error trajectories. Aegies starts with collecting successful, error-free baseline trajectories from a pre-selected set of MAS, then applies targeted LLM-based interventions to the baseline trajectories (by prompt injection and response corruption), and then validates that the system continuing to execute from the corrupted state indeed fails to complete the task, filtering out the ones that still do not lead to a failure. The authors create 9533 annotated error trajectories across 6 MAS with Aegis, and train models using SFT, RL and CL.

**Strengths:**

* Addresses an important problem of data scarcity in training models for MAS error identification.
* Provides a simple intervention to utilize previous successful trajectories.
* Presents a dataset of 9,500+ annotated trajectories, and validate its usefulness by showing OOD generalization in training with this data.

**Weaknesses:**

- Since the technique starts with successful trajectories, it is unclear how this will be applied to MAS built for domains with very high failure rates.
- Due to the dataset being built by corrupting successful trajectories, the dataset will miss out entirely on representing real-world failure modes of the MAS being represented in the dataset.
- Even though an intervention is made in the MAS trajectory to corrupt it, the downstream failure reason could be different from the intervention.

**Questions:**

1. Can the authors comment on how it could be ensured that an error introduced under taxonomy error category X, actually leads to observed failure mode X, and not a different failure mode?
2. Can the authors discuss how Aegis handles representing real-world failure modes of MAS, since it starts from successful trajectories?
3. The studies performed in the paper show improvements in the ability to detect failure modes in a MAS trajectory (through 3 different learning modes), however, it is unclear how this can be utilized to improve the performance of the MAS itself. Could the authors comment on that?

---

> ### Author Response · Authors · 2025-11-20
> **Response to Reviewer Dj9H**
>
> We sincerely appreciate the reviewer’s constructive feedback. We are pleased that you recognized the importance of addressing **data scarcity** in MAS error attribution, the **practicality of our intervention strategy**, and the value of our **large-scale annotated dataset**. These points reflect the core strengths we aimed to contribute. We now address your concerns in detail below.
>
> > W1, W2, Q2: Alignment of Aegis and Real-world Data
>
> Thanks for these insightful questions. As these are central concerns shared by other reviewers, we have provided a comprehensive, multi-perspective answer in our Global Response. To briefly summarize the key points addressing your specific questions:
>
> - Regarding W1 (High Failure Rates): Our approach relies on the established principle of error mode transferability. We do not need successful trajectories for the most complex tasks, but only for a solvable subset. The error signatures learned from these solvable instances transfer to diagnose failures in complex, high-failure domains.
>
> - Regarding W2 & Q2 (Real-world Alignment): Our synthetic failures are not arbitrary; they are programmatic reproductions of the empirical MAST taxonomy, which was derived from real-world failures. Furthermore, our new distributional analysis (Appendix B.3) and OOD correlation results (Appendix D.2) empirically prove that our synthetic data is semantically aligned with and generalizes to real-world failures.
>
> We respectfully refer you to our **Global Response** for the detailed four-part justification and the new experimental evidence supporting these claims.
>
> > W3, Q1: Alignment of introduced error and observed failure mode
>
> We thank you for these insightful and closely related questions (W3 and Q1).  We agree that in complex systems, an initial error can cascade, potentially leading to multiple failure points.  Our framework is explicitly designed to isolate the originating error and ensure our labels are accurate.
>
> Our pipeline's foundation is a set of **deterministic, successful** baseline trajectories.  We first generate a correct reference run by setting agent temperatures to 0 and fixing all random seeds.  The failure is then created by introducing a single, targeted intervention from our adaptive manipulator into this deterministic environment.  By this construction, our injected error is, by definition, the **"earliest decisive error in time."**  It directly aligns with the standard practice for attribution in this field.  As noted in Who&When, when "multiple decisive errors may occur...  the approach is to identify **the earliest error in time as the principal cause of failure**".  Our framework programmatically enforces this principle, guaranteeing that our label corresponds to the principal root cause, not a downstream symptom.
>
> And our manipulator is a context-aware LLM that uses specific, targeted prompts (detailed in **Appendix A.6**) to make the root cause semantically align with the intended MAST error mode.  Our human validation study confirms high label fidelity: programmatic labels achieved an agreement of $\kappa=0.81$ with experts, matching the human-human baseline ($\kappa=0.85$). Please refer to **Point 2 of our Global Response** for details.
>
> > Q3: Improve MAS performance with Aegis
>
> We thank the reviewer for this excellent question about the downstream applications of our work. While the main paper focuses on establishing the diagnostic capability, we agree that its utility in improving MAS performance is a key motivator. To demonstrate this potential, we simulate a simple self-correction loop: when a MAS fails a task, our Aegis-trained model provides an error attribution (faulty agent and error mode) to the MAS, which was then given one retry.
>
> Our initial results show this simple feedback mechanism already provides a positive impact on task success, as seen here:
>
> | MAS | Benchmark | Baseline (Success Rate) | With Aegis-Retry (Success Rate) |
> |---------------|-----------|--------------------------|----------------------------------|
> | Debate        | GSM8K     | 87.8%                   | **89.1%**                        |
> | Debate        | MMLU-pro      | 94.14%                  | **95.04%**                       |
> | Macnet        | GSM8K     | 81.30%                  | **82.78%**                       |
> | Macnet        | MMLU-pro      | 67.21%                  | **69.45%**                       |
> | Magentic-one    | GAIA      | 27.27%                  | **30.13%**                       |
>
> This is just one simple implementation, and we see this as a promising area for future work. We believe the high-fidelity attribution provided by Aegis will be a foundational component for enabling more advanced MAS capabilities, including more sophisticated self-repair, improved agent explainability, and deeper runtime analysis.

---

### Author Response · Authors · 2025-11-20
**Global Responses to All Reviewers**

We thank all reviewers for the time and effort they dedicated to providing this thorough and constructive feedback. To best address the most critical and recurring questions, we provide this global response on **the alignment of our synthetic Aegis data with real-world failure modes**.

We have uploaded a revised version of our paper. All significant additions and modifications to the revised manuscript have been marked **in blue** for easy identification. We believe these additions, supported by new analyses detailed below, will strengthen the paper and address the reviewers' insightful comments.


We are grateful for the opportunity to provide new validating analyses that address this point from four perspectives: **(1) Conceptual Principle, (2) Methodological Grounding, (3) Empirical Data Validation, and (4) Practical Results.**

1. First, on a conceptual principle, our framework is grounded in established academic practice for complex systems diagnosis.  As noted in comprehensive surveys [1], acquiring large-scale labeled data for every new, complex case is "impractical".  This is why the field relies on two key, validated principles that Aegis adopts: (1) error mode transferability, and (2) high-fidelity synthetic data generation.  Critically, the survey also validates our specific generation methodology.  It highlights using "a simulation process...  to explore potential anomalous conditions" and techniques like "adversarial perturbation" as standard, necessary methods to **"mitigate the problem of insufficient samples"**.  Our "correct-to-incorrect" pipeline is a direct implementation of this principle: we use successful trajectories as high-fidelity simulations and apply targeted, context-aware perturbations to programmatically generate the very error data that is otherwise impossible to acquire at scale.

2. Second, our methodology is grounded in real-world data.  Aegis does not invent arbitrary errors.  Our framework programmatically reproduces the 14 known, real-world error modes from the **MAST taxonomy** [2].  This taxonomy was itself empirically derived from a rigorous analysis of over 150 genuine, "naturally erroneous" failure traces.  To validate that our injection of these modes is accurate, we conduct a new human IAA study.  We ask three human annotators (blind to our labels) to classify the root-cause error mode for 100 sampled trajectories.  We find our human-human agreement is $\boldsymbol{\kappa=0.85}$.  Our programmatic Aegis label achieves a program-human agreement of $\boldsymbol{\kappa=0.81}$.  These scores are comparable to the high benchmarks reported in the MAST paper itself ( $\kappa=0.88$ human-human, and $\kappa=0.77$ for their LLM-annotator-human agreement).

3. Third, we perform a three-way semantic comparison against MAST-Data and Who&When. We first extract the semantic content of each trajectory (removing timestamps, metadata, and formatting artifacts) and encode all texts using the *all-MiniLM-L6-v2* model with normalized embeddings. The centroid distances show that Aegis is no farther from a real benchmark than the real benchmarks are from each other:

| Pair | Distance | Type |
| :--- | :--- | :--- |
| Aegis ↔ MAST | 0.521 | Synthetic vs. Real |
| MAST ↔ WhoWhen | 0.490 | Real vs. Real |
| Aegis ↔ WhoWhen | 0.398 | Synthetic vs. Real |

Beyond distances, the clustering metrics (**silhouette = 0.11**) indicate no meaningful separation among the three datasets.  Additionally, the pairwise Mann–Whitney U tests show that the difference between Aegis–MAST (mean **p = 0.097**) is almost identical to that between MAST–Who&When (mean **p = 0.089**), confirming that Aegis differs from real data no more than the real data differ from each other. The corresponding UMAP and t-SNE visualizations are included in **Appendix B.3 (Figure 7)**, and we respectfully invite reviewers to take a look.

4. Finally, the practical results provide definitive proof of generalization. As shown in Table 1, models trained only on our synthetic Aegis data achieve strong OOD performance on the human-annotated Who&When benchmark. To quantify this, we run a new correlation analysis across all models. We find a statistically significant positive correlation between performance on Aegis-Bench and Who&When for the core attribution skills: Agent $\mu$F1 (**r=0.714, p<0.001**) and Error MF1 (**r=0.744, p<0.001**). This proves that training on Aegis successfully teaches models the generalizable skills needed to diagnose real-world failures. The details of the relevant analysis are placed in **Appendix D.2 and Table 11**.

[1] Yan, Peng, et al. "A comprehensive survey of deep transfer learning for anomaly detection in industrial time series: Methods, applications, and directions."

[2] Cemri, Mert, et al. "Why do multi-agent llm systems fail?"

---

### Author Response · Authors · 2025-11-27
**Looking forward to further discussion**

Dear Reviewers,

Thank you again for your thoughtful and constructive feedback on our submission. As the discussion period is entering its final week, we would like to kindly check whether you have any further questions about our rebuttal or the updated analyses in the revised manuscript. All changes are marked in blue for easy reference.

Our response includes new empirical validation, a human annotation study, distributional comparisons with real benchmarks, and additional results demonstrating out-of-distribution generalization. These updates were added to directly address the main concerns raised in the reviews.

We sincerely welcome any continued discussion and are happy to clarify any remaining points. Your feedback has been very valuable in improving the work, and we appreciate your time and engagement.

Best regards,

The Authors

---

### Author Response · Authors · 2025-12-01
**Note to AC: Rebuttal Summary & Acknowledgement**

Dear Area Chair,

First and foremost, we sincerely appreciate you stepping in to oversee our submission under these unprecedented circumstances. We understand the significant additional effort required to evaluate the manuscript and discussion history anew, and we are grateful for your time and dedication to the scientific integrity of the conference.

To assist in your assessment, we summarize below how our revised manuscript and new experiments have addressed the reviewers' concerns, which notably centered on the **alignment between our synthetic Aegis data and real-world failures**.

Since the initial reviews, we have added four new empirical validations to resolve the common concern:

* **Human Validation:** A new study confirms our programmatic labels align with human experts ($\kappa=0.81$), matching human-human agreement ($\kappa=0.85$), validating that our injected errors effectively reproduce real-world failure modes (Addressing **Dj9H**, **nsVH**).
* **Semantic Consistency:** Embedding analyses demonstrate that Aegis trajectories are distributionally close to real-world benchmarks like MAST and Who&When (Addressing **HRKG**, **nsVH**).
* **OOD Generalization:** We found a strong, statistically significant correlation ($r=0.744, p<0.001$) between model performance on Aegis and the real-world Who&When benchmark (Addressing **HRKG**).
* **Downstream Utility:** New experiments show that Aegis-trained models successfully improve MAS pass rates via a simple self-correction loop (Addressing **Dj9H**).

**Addressed Specific Technical Concerns:**
* **Causal Validity:** Clarified the "earliest decisive error" logic to ensure label accuracy (Response to **Dj9H**).
* **Model Details:** Added the DCL algorithm details and qualitative error analysis case studies (Response to **nsVH**).
* **Taxonomy Origin:** Clarified our use of the empirically-grounded MAST taxonomy (Response to **HRKG**).

We believe these updates resolve the reservations of Reviewers Dj9H, nsVH, and HRKG, complementing the positive assessment from Reviewer qv7V. We hope this overview assists you in your final assessment.

Best regards,

The Authors

---

### Meta-Review · Area_Chair_Dz9K · 2026-01-06

**Summary:**

The paper presents Aegis, an automated framework to programatically generate large-scale error attribution data in multi-agent LLMs. The key idea is to inject controlled, context-aware faults into correct trajectories collected from pre-defined multi-agent systems. Aegis constructs a large dataset of 9,533 trajectories with annotated failure attirubtions, which is used to train models to improve attribution for different learning paradigms including Supervised Fine-Tuning, Reinforcement Learning, and Contrastive Learning.

The reviewers all appreciated the scalability of the pipeline, the principled approach to data generation, and the strong Out-of-Distribution (OOD) performance on real-world benchmarks like Who&When. The concerns in the initial reviews are mainly about gap between the synthetic data and real-world error, where the errors injected into correct paths accurately represent natural failures. The authors provided extensive new results during the rebuttal on alignment of Aegis with real-world data, which AC believes that it addressed the concerns. The paper is a timely contribution to the community and could be a useful resource for future research on multi-agent LLMs.

**Reviewer Concerns:**

Addressed Concerns:

Alignment with real-world data: shared by Reviewers Dj9H, nsVH, and HRkG. The authors provided extensive new results during the rebuttal: a human validation study showing high agreement with experts, semantic consistency checks (UMAP/t-SNE) showing Aegis trajectories overlap with real-world data, and a statistically significant correlation between Aegis and Who&When.

Technical clarity: Concerns regarding the DCL algorithm details and taxonomy origin were resolved.

Outstanding Concerns:

No obvious remaining concerns.

**Reviewer Scores:**

- Reviewer qv7V (8 -> 8): I believe the reviewer will remain positive.

- Reviewer HRkG (6 -> 6/8), Reviewer nsVH (6 -> 6/8): Likely to remain positive or even increase their score given the added analysis on alignment with real-world data.

- Reviewer Dj9H (4 -> 6): The human validation and downstream utility experiments addressed the reviewer's primary concerns.

---

### Decision · Program_Chairs · 2026-01-26

Accept (Poster)